# Ultrasonic Detection Method for Grouted Defects in Grouted Splice Sleeve Connector Based on Wavelet Pack Energy

**DOI:** 10.3390/s19071642

**Published:** 2019-04-06

**Authors:** Zuohua Li, Lilin Zheng, Chaojun Chen, Zhili Long, Ying Wang

**Affiliations:** 1School of Civil and Environment Engineering, Harbin Institute of Technology, Shenzhen 518055, China; lizuohua@hit.edu.cn (Z.L.); lilinz@foxmail.com (L.Z.); 2School of Mechanical Engineering and Automation, Harbin Institute of Technology, Shenzhen 518055, China; longzhili@hit.edu.cn; 3Department of Civil and Environmental Engineering, University of Surrey, Guildford GU2 7XH, UK; ying.wang@surrey.ac.uk

**Keywords:** grouted splice sleeve connector, precast concrete structure, ultrasonic detection method, wavelet packet energy, grouted defect

## Abstract

Grouted splice sleeve (GSS) connectors are mainly used in precast concrete structures. However, errors in manual operation during construction cause grouted defects in the GSS connector, which can lead to a negative effect on the overall mechanical properties of the structures. Owing to the complex structure of precast concrete members with a GSS connector, it is difficult to detect grouted defects effectively using traditional ultrasonic parameters. In this paper, a wavelet packet analysis algorithm was developed to effectively detect grouted defects using the ultrasonic method, and a verified experiment was carried out. Laboratory detection was performed on the concrete specimens with a GSS connector before grouting, in which the grouted defects were mimicked with five sizes in five GSS connectors of each specimen group. A simple and convenient ultrasonic detection system was developed, and the specimens were detected. According to the proposed grouted defect index, the results demonstrated that when the grouted defects reached certain sizes, the proposed method could detect the grouted defects effectively. The proposed method is effective and easy to implement at a construction site with simple instruments, and so provides an innovative method for grouted defects detection of precast concrete members.

## 1. Introduction

Precast concrete structures have received increasing research interest recently, especially in a global trend of off-site modular construction. Their advantages include: effective control over the quality of materials and workmanship, low resource consumption and cost efficiency, besides, the on-site construction/assembly of precast concrete structures benefits from the low impact of weather conditions and low labor demand [1,2,3]. The most important part used to connect precast concrete members is called the grouted splice sleeve (GSS) connector, which is composed of a splice sleeve, connecting a steel rebar and high-strength, micro-expansion, cement-based grouting materials. In order to ensure the continuity of load transfer, grouting materials need to be viewed as high performance bonding materials for the connecting steel rebar [4]. The grouted defect in a GSS connector remains a significant concern, because the grouted defect due to grout operation mistakes can negatively affect load transfer and decrease the bearing capacity and seismic performance of the precast structures.

Owing to the complex structure of precast concrete members with a GSS connector and the concealment of grouting material inside the splice sleeve, it is quite difficult to detect grouted defects inside the splice sleeve. Current research mainly focuses on the bond anchorage mechanism [5,6], joint mechanical properties [7,8], and deformation or damage monitoring [9,10], but there is limited research on the detection of grouted defects in the GSS connector. Although an industrial computed tomography (CT) can achieve the detection of grouted defects in a GSS connector in the laboratory [11,12], due to the high hardware requirements, high cost, and harsh application conditions, industrial CT instruments are currently limited to laboratory application. Therefore, it is a challenging research topic to develop a grouted defects detection method which is reliable and easy to implement on site.

Alternatively, non-destructive testing methods such as the impact-echo method [13], ground-penetrating radar method [14], infrared thermal image method [15], and ultrasonic method [16], are convenient for outdoor field application. Among them, the ultrasound method has been widely used due to its low hardware requirements, excellent directivity, strong penetration ability and abundant information. Many scholars have carried out a series of research on the ultrasonic method regarding concrete strength [17], concrete defect [18], concrete thickness [19], damage layer thickness of concrete [20] and its influencing factors [21]. However, complex materials such as concrete-steel interfaces, and noise, and waveform distortion lead to difficulties in the interpretation of ultrasonic sound signals, misjudgment of the first wave amplitude, waveform and frequency. Therefore, it is difficult to identify grouted defects in a GSS connector effectively using the traditional ultrasonic parameters.

With the development of signal processing technology, more powerful parameters of ultrasonic detection have been developed for the defect detection from the energy perspective [22]. In Ref. [23], a structural health monitoring method based on the wavelet packet was proposed, and the effective monitoring of the generation and development process of cracks under different damage states of members by using piezoelectric aggregates was established. In Ref. [24], a method based on the energy spectrum analysis of the piezoelectric ceramic stress wave wavelet packet to identify the interfacial bond performance of a concrete-filled steel tubular column was proposed, and the result showed that the method could achieve relatively ideal detection. In Ref. [25], the piezoelectric aggregate excitation stress wave was used to detect the damage of a simulated concrete beam. These results showed that the damage index based on the wavelet packet analysis was very sensitive to the damage of members within a certain propagation distance and it could effectively reflect the damage degree of members. Therefore, the wavelet packet analysis algorithm, which has significant potential in the detection of grouted defects in a GSS connector, can be adopted to analyze the ultrasonic detection signal of precast concrete members.

In this paper, the wavelet packet analysis algorithm was proposed as a new detection method for grouted defects using ultrasonic methods. A grouted defect indexes (GDI) was established to quantify the degree of grouted defects. Then an ultrasonic detecting system was developed using an ultrasonic generator, power amplifier, oscilloscope, ultrasonic probes, and other instruments. Finally, four different groups of concrete specimens with a GSS connector containing defects were constructed and tested by the proposed method and the development system.

## 2. Fabrication of Concrete Specimens with a GSS Connector

The specimens studied were four different groups of concrete specimens with a GSS connector. Specifically, the groups SJ1 and SJ2 were made of concrete and a centered GSS connector. The group SJ3 was made of concrete, a centered GSS connector and longitudinal rebar on both sides. The group SJ4 was made of concrete, an offset GSS connector, and a longitudinal rebar on only one side.

The artificial grouted defects of concrete specimens were simplified and set as circumferential defects along the connecting rebar. The sizes and the defect settings of these specimens are listed in Table 1. Their schematic diagrams are shown in Figure 1, and their physical diagrams are shown in Figure 2.

## 3. Methodology

In this paper, the wavelet packet analysis was used as a signal processing tool to analyze the ultrasonic detection signal of precast concrete members with a GSS connector, and an evaluation index based on the wavelet packet energy was proposed and employed to detect the grouted defect in the GSS connector.

### 3.1. Wavelet Packet Analysis

Wavelet analysis can be viewed as an extension of the Fourier transform. The Fourier transform decomposes a signal into sine waves with different frequencies and phases. Similarly, wavelet analysis decomposes a signal into shifted and scaled sub-signals of the mother wavelet. Wavelet is a finite duration waveform, and its average value is zero.
(1)$$\int_{-\infty }^{+\infty }\psi (t)dt=0$$
where ψ(t) is a selected mother wavelet function.

The continuous wavelet transfer of a signal f(t) is defined as
(2)$$W_{f}(a,b)=\frac{1}{\sqrt{a}}\int_{-\infty }^{+\infty }f(t)\bar{\psi }(\frac{t-b}{a})dt$$
where *a* > 0 is a dilation parameter and b∈R is a translation parameter.

The scaling function of wavelet analysis varies by binary system. The result of the time-frequency decomposition of the detection signal is the frequency sub-bands divided exponentially at equal intervals. Therefore, the frequency resolution at high frequency sub-bands and the temporal resolution at low frequency sub-bands are all inferior. However, the wavelet packet analysis decomposes both the high frequency part and the low frequency part according to the characteristics of an ultrasonic detection signal, adaptively selects the corresponding frequency sub-bands, and distributes different signal components to different frequency sub-bands. Thus, the time-frequency resolution can be improved. Wavelet packet analysis solves the shortcomings of wavelet analysis, and it has more extensive application value [26,27].

In multi-resolution analysis, the Hilbert space $L^{2}(R)$ is decomposed into the orthogonal sum of all subspace $W_{j}(j\in Z)$ according to the different scaling factors *j*. Wavelet packet analysis subdivides the wavelet subspace $W_{j}$ further to improve the high frequency resolution of a signal. The wavelet subspace $W_{j}$ and scaling space $V_{j}$ are characterized by a new subspace $U_{j}^{n}$.
(3)$$\{\begin{array}{c} U_{j}^{0}=V_{j},j\in Z\\ U_{j}^{1}=W_{j},j\in Z \end{array}$$

Then the orthogonal decomposition $V_{j+1}=V_{j}⊕W_{j}$ of the Hilbert space can be unified as Equation (4) through the decomposition of $U_{j}^{n}$.
(4)$$U_{j+1}^{0}=U_{j}^{0}⊕U_{j}^{1},j\in Z$$

The subspace $U_{j}^{n}$ is defined as the closure space of function $u_{n}(t)$, and $U_{j}^{2n}$ is defined as the closure space of function $u_{2n}(t)$. Let $u_{n}(t)$ satisfy the following two-scale equation
(5)$$\{\begin{array}{c} u_{2n}(t)=\sqrt{2}\sum_{k\in Z}h(k)u_{n}(2t-k)\\ u_{2n+1}(t)=\sqrt{2}\sum_{k\in Z}g(k)u_{n}(2t-k) \end{array}$$
where $g(k)={\left(-1\right)}^{k}h(1-k)$, meaning that the two coefficients are orthogonal.

When *n* = 0
(6)$$\{\begin{array}{c} u_{0}(t)=\sum_{k\in Z}h_{k}u_{0}(2t-k)\\ u_{1}(t)=\sum_{k\in Z}g_{k}u_{0}(2t-k) \end{array}$$

In multi-resolution analysis, scale function ϕ(t) and wavelet function ψ(t) satisfy the following two-scale equation.
(7)$$\{\begin{array}{c} \phi (t)=\sum_{k\in Z}h_{k}\phi (2t-k),{\left\{h_{k}\right\}}_{k\in Z}\in l^{2}\\ \psi (t)=\sum_{k\in Z}g_{k}\phi (2t-k),{\left\{g_{k}\right\}}_{k\in Z}\in l^{2} \end{array}$$

From Equations (6) and (7), $u_{0}(t)$ degenerates to scale function ϕ(t), and $u_{1}(t)$ degenerates to wavelet function ψ(t).

Equations (4) and (6) are equivalent. If the representation is extended to the case of $n\in Z_{+}$, and then
(8)$$U_{j+1}^{n}=U_{j}^{n}⊕U_{j}^{2n+1},j\in Z;n\in Z_{+}$$

Let $g_{j}^{n}(t)\in U_{j}^{n}$, $g_{j}^{n}$ can be expressed as
(9)$$g_{j}^{n}(t)=\sum_{l}d_{l}^{j,n}u_{n}(2^{j}t-l)$$

Then, the wavelet packet decomposition algorithm is
(10)$$\{\begin{array}{c} d_{l}^{j,2n}=\sum_{k}a_{k-2l}d_{k}^{j+1,n}\\ d_{l}^{j,2n+1}=\sum_{k}b_{k-2l}d_{k}^{j+1,n} \end{array}$$

More details about above derivations can be found from Refs. [28,29,30].

### 3.2. Proposed Grouted Defect Index

In this paper, an evaluation index is defined based on the ultrasonic detection signal processed by the wavelet packet analysis. Suppose that an ultrasonic detection signal *S* is decomposed by an *N*-level wavelet packet decomposition into a 2*^N^* signal set {*X*_1_, *X*_2_, *X*_3_, …, *X*_2_*^N^*} with
(11)$$X_{j}=[x_{j,1},x_{j,2},x_{j,3},\cdots ,x_{j,m}]$$
where *m* is the amount of sampling data and *j* is the sequence number of sub-bands.

And *E_i, j_* is the energy of the decomposed signal which can be described as
(12)$$E_{i,j}={\left\|X_{j}\right\|}^{2}=x_{j,1}^{2}+x_{j,2}^{2}+x_{j,3}^{2}+\cdots +x_{j,m}^{2}$$

Then the wavelet packet energy of the signal *S* obtained under different scenarios is defined as
(13)$$E_{i}=[E_{i,1},E_{i,2},E_{i,3},\cdots ,E_{i,2^{N}}]$$

The wavelet packet energy of each sub-band in the main frequency range of an excitation signal is $E_{i,k},E_{i,k+1},E_{i,k+2},\cdots ,E_{i,k+l}$, then the wavelet packet energy vector (WPEV) is defined as
(14)$$ER_{i}=[e_{i,1},e_{i,2},e_{i,3},\cdots ,e_{i,n}]$$
where $e_{i,1}=\frac{E_{i,k}}{EP_{i}},e_{i,2}=\frac{E_{i,k+1}}{EP_{i}},e_{i,3}=\frac{E_{i,k+2}}{EP_{i}},\cdots ,e_{i,n}=\frac{E_{i,k+l}}{EP_{i}}$.

*EP_i_* is the sum of the wavelet packet energy of sub-bands in the main frequency range of the excitation signal.
(15)$$EP_{i}=E_{i,k}+E_{i,k+1}+E_{i,k+2}+\cdots +E_{i,k+l}$$

The WPEV of an ultrasonic detection signal obtained from a GSS connector without grouted defect is defined as $ER_{h}=[e_{h,1},e_{h,2},e_{h,3},\cdots ,e_{h,n}]$, and the WPEV of an ultrasonic detection signal obtained from a GSS connector with grouted defects is defined as $ER_{i}=[e_{i,1},e_{i,2},e_{i,3},\cdots ,e_{i,n}]$. Based on the proposed method, a grouted defect index (GDI) which is used to judge the existence of grouted defects can be defined as
(16)$$GDI=\sqrt{{\sum_{u=1}^{n}\left(\frac{e_{i,u}-e_{h,u}}{e_{h,u}}\right)}^{2}}$$

The definition of GDI filters the sub-bands obtained by the wavelet packet decomposition. The interference of clutter can be better eliminated, and the difference between the grouted defect detection signals and grouted compactness detection signals is highlighted. It is clear that the defined GDI indicates the change of ultrasonic transmission energy caused by the grouted defects. When the value of GDI is greater than a certain value, a grouted defect exists in the member.

Symlets wavelet base sym14 was regarded as the mother wavelet. The frequency band was not overlapped because of the orthogonality of Symlets wavelet base. The ultrasonic detection signals of the specimens were decomposed into ten levels. The wavelet packet coefficients of the 10th level were extracted to calculate the energy proportion of frequency sub-bands 3, 4, 5 and 6 (the frequency range was from 24.42 to 73.26 kHz), and then the WPEVs of the detection signals were built. To reduce the influence of the operation error, the average of the three groups’ WPEVs at each point was taken as its final WPEV. Finally, the GDIs of each specimen were calculated using the proposed Equation (16).

## 4. Experimental Development

A simple and convenient detecting system was designed to detect the grouted defects. A schematic diagram and a picture of the system are shown in Figure 3 and Figure 4, respectively. The system contained four components. Specifically, ① an ultrasonic generator (33511B; Agilent, CA, USA); ② a power amplifier (ATA-1200; Antai, Shaanxi, China); ③ an oscilloscope (MDO3024; Tektronix, OR, USA), and its sampling rate was 25 MSa/s in all experiments; and ④ two ultrasonic probes for longitudinal wave generation (SIUI, Guangdong, China; central frequency: 50 kHz). In addition, a water-based polymer gel ultrasound coupler was used to effectively couple the probes to the surface of the concrete specimens.

How to accurately collect the ultrasonic detection signals was the main objective of the defect detecting system. In this paper, the ultrasonic generator generated an excitation signal, a five-peaks sinusoidal signal modulated by the Hanning window, which could be transmitted into the oscilloscope and displayed in channel one. Then, a longitudinal wave would be generated after the signal went through the probe. The longitudinal wave incidented from one side of the specimen propagating to the other side and could be obtained by the receiving probe. After the receiving probe collected the ultrasonic detection signal, the detection signal was transmitted to the oscilloscope and displayed in channel two.

Before detection, it was necessary to deal with the surface of these specimens properly and to arrange the detecting lines in combination with the internal defects of the specimens. After surface treatment, the detecting line of each group was arranged as Figure 5. Importantly, a detecting line was laid directly above and below the sleeve center line of each specimen. The SJ1 group was equipped with five detecting points on each line, which meant that there were five pairs of detecting points on each specimen. The SJ2, SJ3 and SJ4 groups of specimens were equipped with six detecting points on each line. The naming method of each detecting point was group number-specimen number-detecting point number. For example, SJ111 represents detecting point 1 of the first specimen in the SJ1 group.

## 5. Experimental Results and Analysis

### 5.1. The WPEVs of Ultrasonic Detection Signals of Detecting Points

Four groups of specimens with 115 detecting points were detected in the experiment. The WPEVs of each detecting point in each specimen are shown in Figure 6, Figure 7, Figure 8 and Figure 9. From Figure 6, Figure 7, Figure 8 and Figure 9, the WPEVs at different points on the same specimen were different. The reason was that the random distribution of aggregates affected the ultrasound propagation in a specimen. When the ultrasound propagated, reflecting and scattering waves would be produced on the aggregate surface, and the ultrasonic energy would change with time. The particle sizes and locations of the aggregates in concrete were random. As a result, the aggregate distribution of the different detecting points was slightly different, which led to the difference of the WPEVs of the ultrasonic detection signal under the same grouted defect setting.

The WPEVs of point one on each specimen was used as a representative to be analyzed. Thus, from Figure 6a, Figure 7a, Figure 8a and Figure 9a, firstly the energy of ultrasonic detection signals was mainly concentrated in frequency sub-band four (the frequency range was from 36.63 to 48.84 kHz) and five (the frequency range was from 48.84 to 61.05 kHz). The sum of the energy proportions of these two sub-bands exceeded 0.8, which was higher than that of sub-band three (the frequency range was from 24.42 to 36.63 kHz) and six (the frequency range was from 61.05 to 73.26 kHz). Secondly, the wavelet packet energy in the specified frequency sub-bands of different defect widths in one group was changed. In the SJ1 group, SJ2 group and SJ4, it was obvious that the difference of the WPEVs between the specimen without grouted defect and specimens with grouted defects was at a maximum in frequency sub-bands four and five, specifically, in frequency sub-band four in the SJ3 group. Thirdly, in the SJ1 group, SJ2 group and SJ3 group, when the grouted defects were less than 35 mm, the difference of the WPEV between the specimen without grouted defects and the specimens with grouted defects was not obvious. When the defects were more than 37 mm, it became obvious. In other words, when the grouted defects of the SJ4 group were less than 40 mm, the difference of WPEVs between the specimen without grouted defect and the specimens with grouted defects were not obvious, and when the defect reached 46.5 mm, it immediately became obvious.

The ultrasound detection signal of SJ445 was abnormal. The phenomenon was due to the slight protrusion of aggregate on the specimen surface, which made the transducer unable to better fit the surface of the component and affect the incidence of the ultrasound. Thus, the point needed to be eliminated.

Analysis of frequency sub-bands indicated that the grouted defect in a GSS connector could have a significant impact on the signal energy distribution. When the ultrasonic wave encountered grouted defects in the propagation, diffraction, and scattering occurred, and the energy mainly changed at frequency sub-band four and five. From the perspective of energy variation, it could be seen that only when the grouted defects reached certain sizes would it cause the obvious change of the energy of ultrasonic detection signal, and the energy proportion in different frequency sub-bands had a significant difference, which was presented as the difference of the WPEVs between the specimen without grouted defects and the specimens with grouted defects. Consequently, it could be concluded that the WPEV of a detection signal could effectively reflect the grouted defect in a GSS connector.

### 5.2. Grouted Defects Detection Using Wavelet Packet Analysis-Based Index

From the point of energy variation, the proposed grouted defect index (GDI) was used to evaluate the energy level of ultrasonic detection signal of specimens with different sizes of grouted defects, so as to identify the grouted defects.

The GDIs of the detecting points on the specimens without grouted defects was calculated based on the WPEV of a detecting point on the SJ11, SJ21, SJ31 and SJ41 respectively, where the results are shown in Figure 10.

Owing to the influence of concrete aggregates distribution, the WPEVs of the ultrasonic detection signals of different points on a specimen were different, and this led to its GDIs fluctuating within a certain range. As shown in Figure 10, the GDI of each specimen without grouted defects was less than 0.14. Thus, the grouted defect identification baseline in the experiment was 0.14, which meant that when all the GDIs of a specimen with certain size of grouted defect was more than 0.14, it was considered that this grouted defect could be effectively distinguished by the GDI.

The GDIs of different specimen groups are shown in Figure 11, Figure 12, Figure 13 and Figure 14.

As shown in Figure 11, for the SJ1 group, when the widths were 27 mm (SJ12) and 32 mm (SJ13), the GDIs of several detecting points were more than 0.14. When the widths were 37 mm (SJ14) and 42.5 mm (SJ15), all the GDIs of the detecting points were more than 0.14. The grouted defects of 27 mm and 32 mm could not be effectively identified by GDI, and the grouted defects of 37 mm and 42.5 mm could be effectively identified.

Similarly, from Figure 12, Figure 13 and Figure 14, for the SJ2 group, the grouted defects of 30 mm (SJ22) and 35 mm (SJ23) could not be effectively identified by GDI, and the grouted defects of 40 mm (SJ24) and 46.5 mm (SJ25) can be effectively identified. For the SJ3 group, the grouted defects of 30 mm (SJ32), 35 mm (SJ33) and 40 mm (SJ34) could not be effectively identified by GDI, and the grouted defects of 46.5 mm (SJ35) could be effectively identified. For SJ4 group, the grouted defects of 30 mm (SJ42) and 35 mm (SJ43) could not be effectively identified by GDI, and the grouted defects of 40 mm (SJ44) and 46.5 mm (SJ45) could be effectively identified.

In summary, the proposed ultrasonic detection method based on wavelet packet energy could effectively identify the grouted defects which were more than 37 mm in the SJ1 and SJ2 group specimens, 46.5 mm in the SJ3 group specimens, and more than 40 mm in the SJ4 group specimens.

In practice, several typical precast concrete members’ models could be prepared in the laboratory according to the construction process and materials at the construction site. The grouting operation was carried out carefully according to the requirement to ensure that there was no grouted defect in models. The models were detected according to the proposed method, and the WPEVs and GDIs were calculated to obtain the grouted defect identification baseline. Meanwhile, the precast concrete members at the construction site were also detected, and then the GDIs of members were calculated based on the WPEVs obtained from the laboratory models. Finally, compared with the obtained identification baseline, if the GDI of a member exceeds the line, it can be regarded that the grouted defect exists in the member.

## 6. Conclusions

In this paper, an ultrasonic detection method based on wavelet packet energy for grouted defects in a GSS connector of the precast concrete structure and an index to judge the existence of the grouted defects was proposed. The effectiveness of the proposed method was experimentally verified with a simple and convenient detection system and four groups of concrete specimens with GSS connectors. Based on the above results, the following conclusions can be made.
The grouted defects reach certain widths, which leads to the obvious difference of the WPEVs between the specimen without grouted defects and the specimens with grouted defects.The WPEVs of the ultrasonic detection signals of different points on a specimen are different because of the influence of concrete aggregates distribution.Under an excitation with 50 kHz, the grouted defect identification baseline of specimens with a GSS connector is 0.14.The method can effectively detect grouted defects with certain widths. Specifically, for the specimens made of concrete and a centered GSS connector, their specific grouted defects (the widths are more than 37 mm) can be effectively detected. For the specimens made of concrete, a centered GSS connector, and longitudinal rebars on both the two sides, the grouted defects with a 46.5 mm width can be effectively detected. For the specimens made of concrete, an offset GSS connector, and a longitudinal rebar on only one side, their specific grouted defects (the widths are more than 40 mm) can be effectively detected.

The proposed ultrasonic detection method based on wavelet pack energy achieved effective detection of grouted defects. Moreover, it has low instrument requirement, simple application conditions, and it is easy to apply at the construction site, which provides good potential in the detection of grouted defects of precast concrete members. In order to better promote the method in engineering practice, follow-up research can be conducted from the following aspects.
There are transverse rebars in precast concrete members, which will affect the detection. The influence of transverse rebars can be further studied.The environmental factors, such as temperature and humidity, are different in different areas. The influence of environmental factors can be further studied to adapt the method to the construction sites located in various areas.

## Figures and Tables

**Figure 1 sensors-19-01642-f001:**
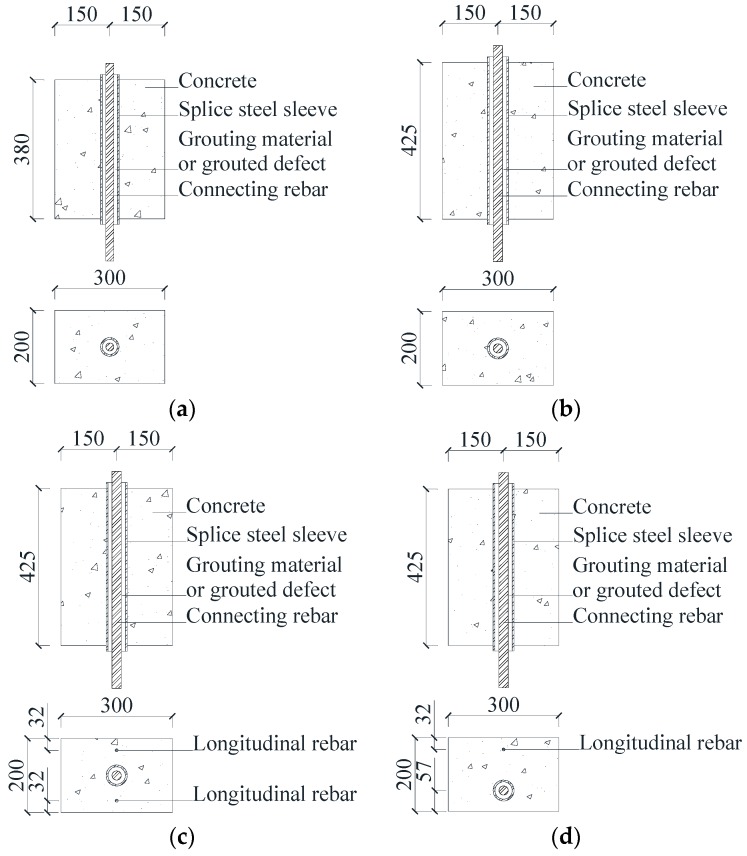
The schematic diagram of the concrete specimens. (**a**) SJ1 group, (**b**) SJ2 group, (**c**) SJ3 group, (**d**) SJ4 group.

**Figure 2 sensors-19-01642-f002:**
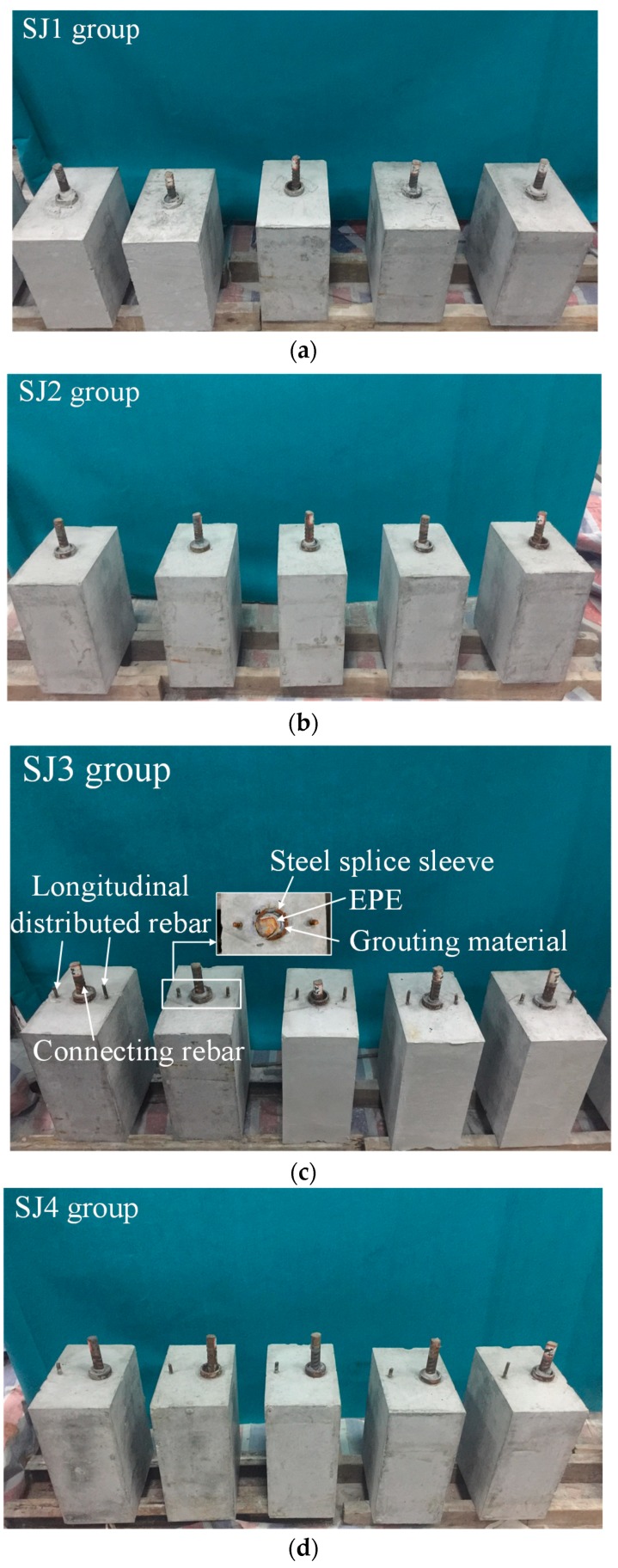
The physical diagram of the concrete specimens. (**a**) SJ1 group, (**b**) SJ2 group, (**c**) SJ3 group, (**d**) SJ4 group.

**Figure 3 sensors-19-01642-f003:**
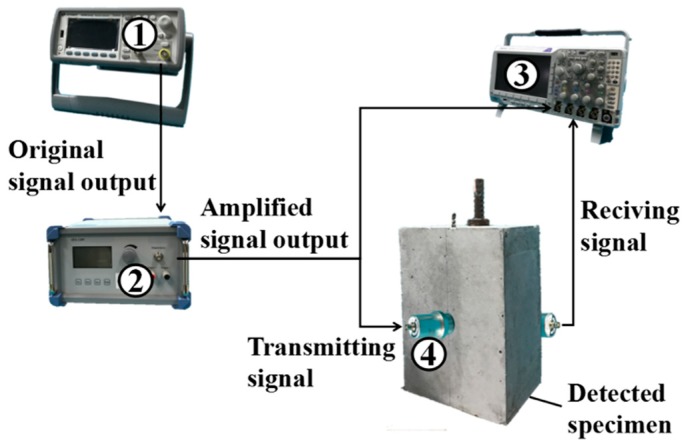
The schematic diagram of the detecting system.

**Figure 4 sensors-19-01642-f004:**
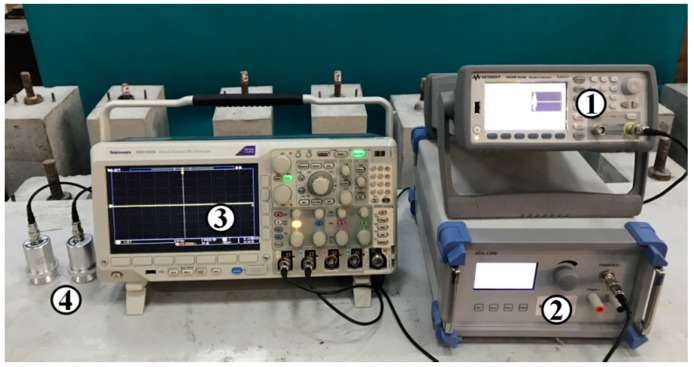
The physical diagram of the detecting system.

**Figure 5 sensors-19-01642-f005:**
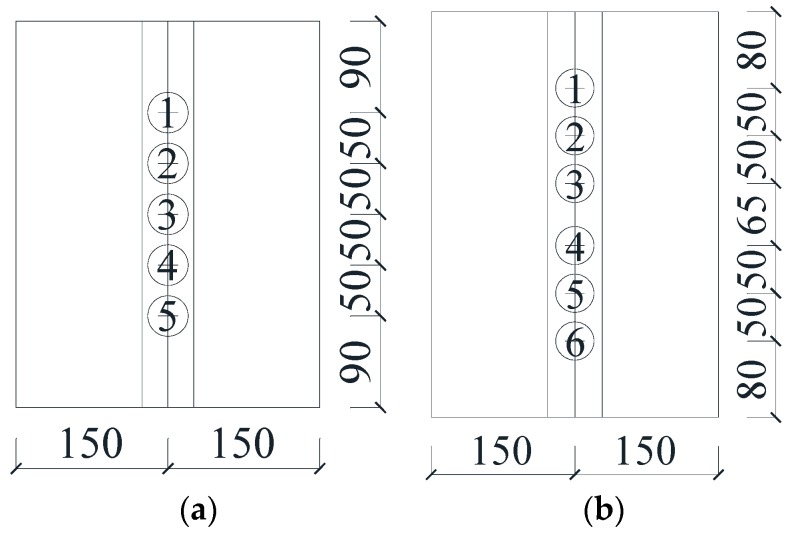
The schematic diagram of the different detecting points. (**a**) Detecting points of SJ1 group, (**b**) Detecting points of SJ2, SJ3, and SJ4 group.

**Figure 6 sensors-19-01642-f006:**
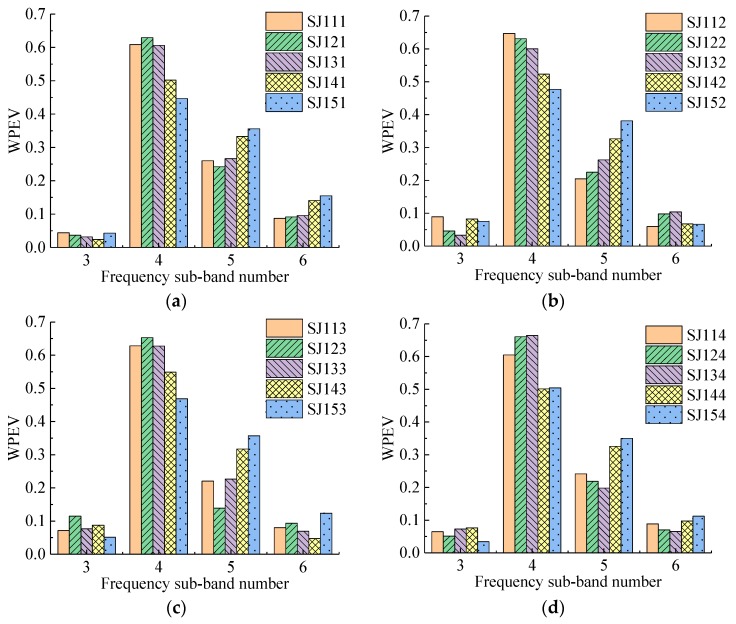
The wavelet packet energy vectors (WPEVs) of different detecting points of SJ1 group. (**a**) Detecting point 1, (**b**) Detecting point 2, (**c**) Detecting point 3, (**d**) Detecting point 4, (**e**) Detecting point 5.

**Figure 7 sensors-19-01642-f007:**
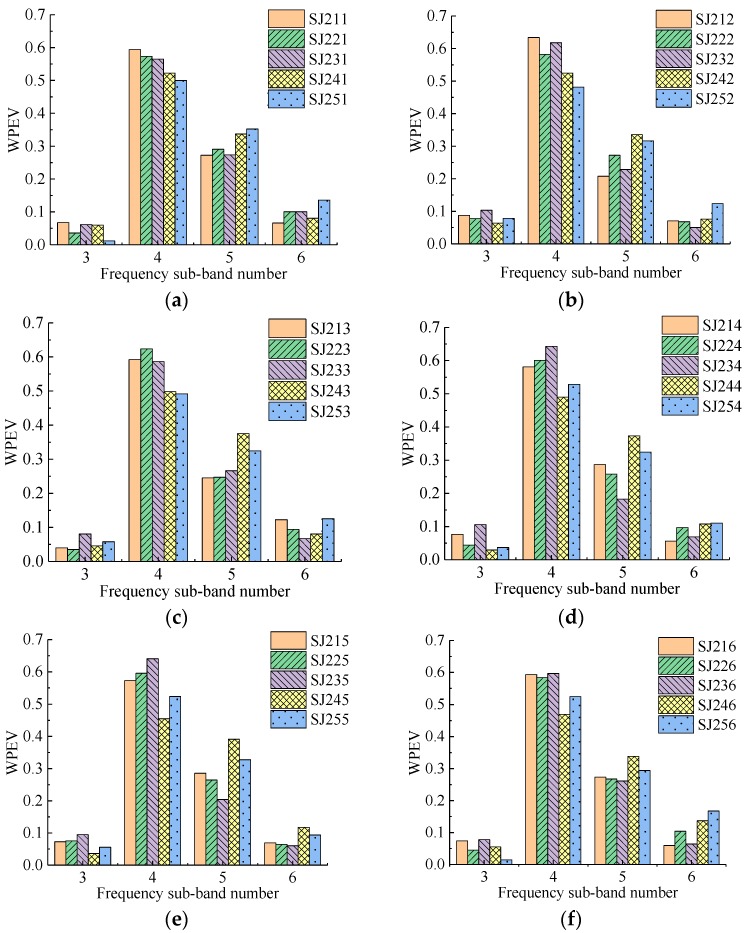
The WPEVs of different detecting points of the SJ2 group. (**a**) Detecting point 1, (**b**) Detecting point 2, (**c**) Detecting point 3, (**d**) Detecting point 4, (**e**) Detecting point 5, (**f**) Detecting point 6.

**Figure 8 sensors-19-01642-f008:**
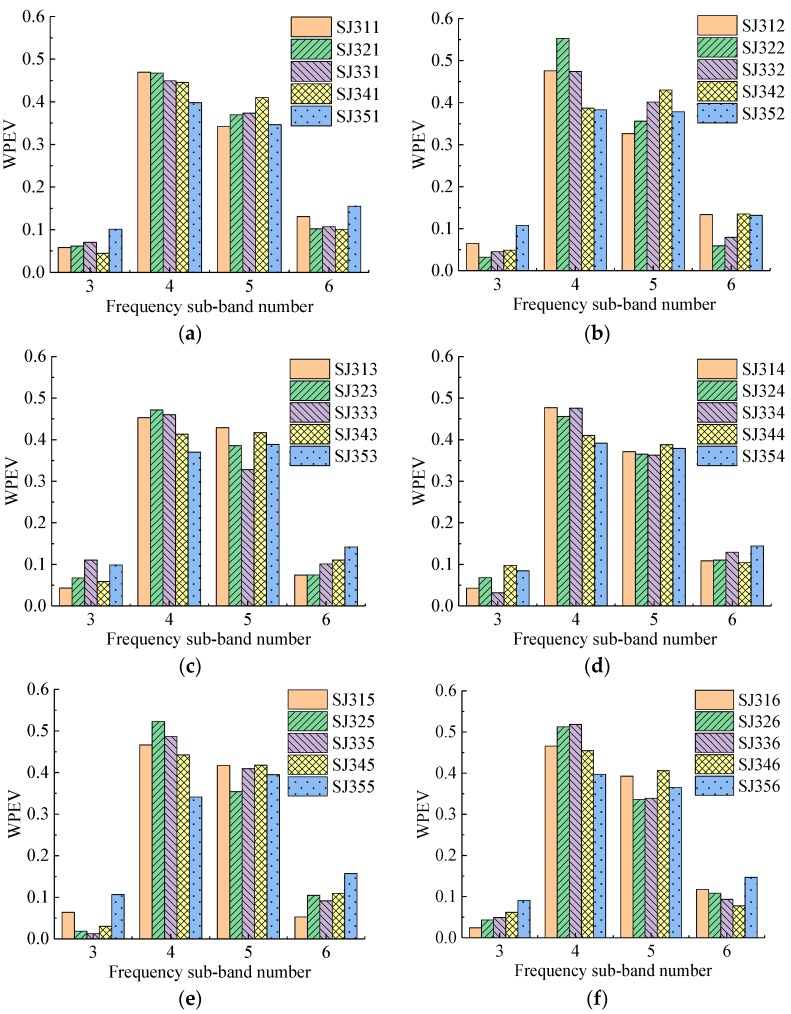
The WPEVs of different detecting points of the SJ3 group. (**a**) Detecting point 1, (**b**) Detecting point 2, (**c**) Detecting point 3, (**d**) Detecting point 4, (**e**) Detecting point 5, (**f**) Detecting point 6.

**Figure 9 sensors-19-01642-f009:**
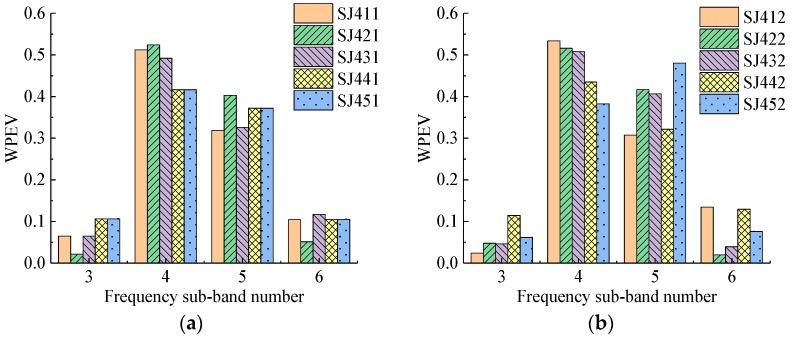
The WPEVs of different detecting points of the SJ4 group. (**a**) Detecting point 1, (**b**) Detecting point 2, (**c**) Detecting point 3, (**d**) Detecting point 4, (**e**) Detecting point 5, (**f**) Detecting point 6.

**Figure 10 sensors-19-01642-f010:**
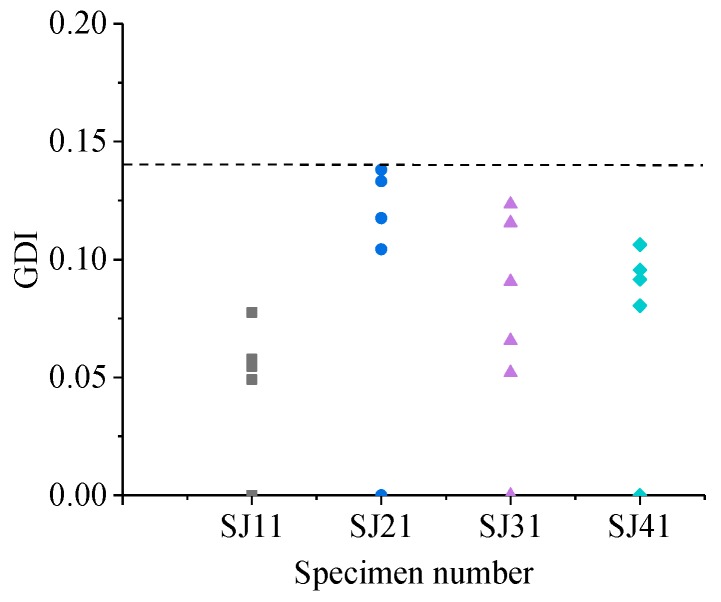
Grouted defect index (GDI) of specimens without grouted defect.

**Figure 11 sensors-19-01642-f011:**
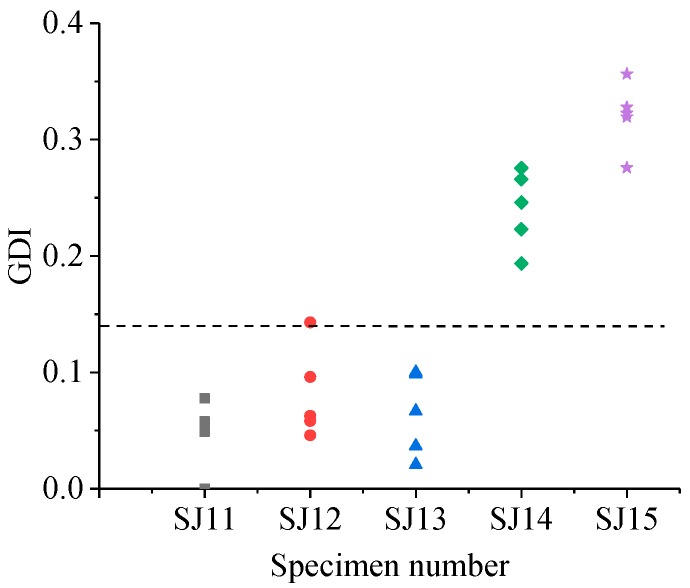
GDI of specimens in the SJ1 group.

**Figure 12 sensors-19-01642-f012:**
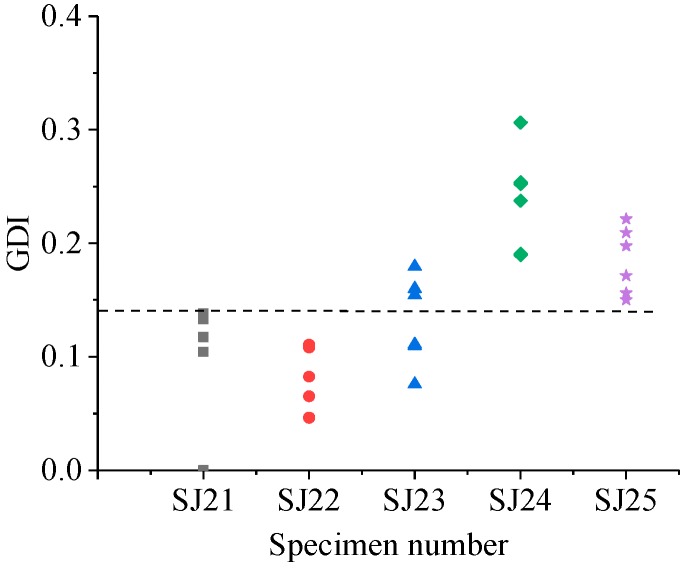
GDI of specimens in the SJ2 group.

**Figure 13 sensors-19-01642-f013:**
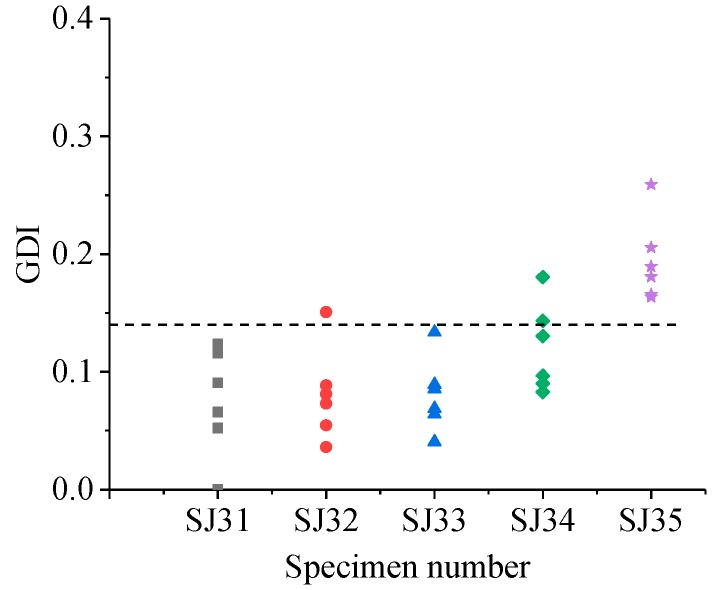
GDI of specimens in the SJ3 group.

**Figure 14 sensors-19-01642-f014:**
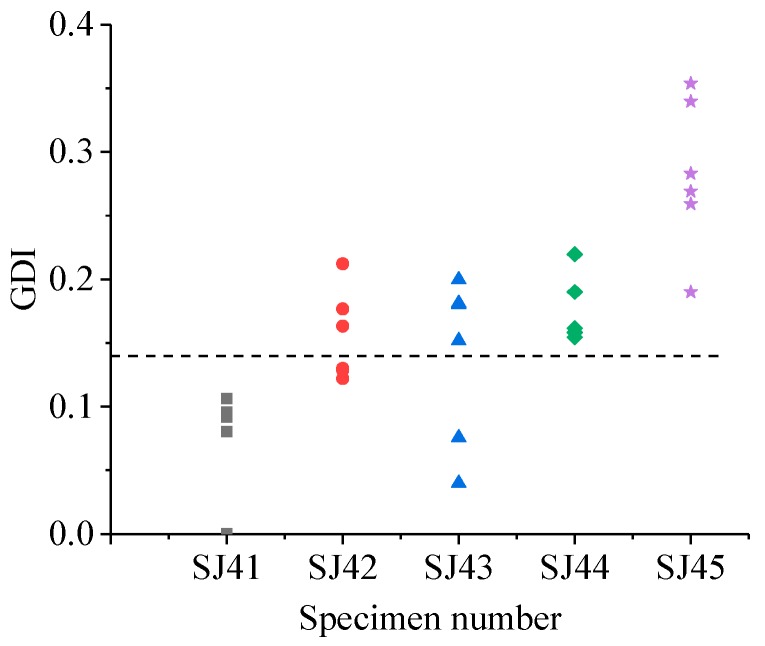
GDI of specimens in the SJ4 group.

**Table 1 sensors-19-01642-t001:** The sizes and the defect settings of the specimens.

Specimens Number	Sleeve Type	Sleeve Inside/ Outside Diameter (mm)	Specimens Size (mm)	GSS Connector Location	Grouted Defect Width (mm)
SJ11	GT22	42.5/54	200 × 300 × 380	Centered	0
SJ12					27
SJ13					32
SJ14					37
SJ15					42.5
SJ21	GT25	46.5/58	200 × 300 × 425	Centered	0
SJ22					30
SJ23					35
SJ24					40
SJ25					46.5
SJ31	GT25	46.5/58	200 × 300 × 425	Centered and two rebars	0
SJ32					30
SJ33					35
SJ34					40
SJ35					46.5
SJ41	GT25	46.5/58	200 × 300 × 425	Offset and one rebar	0
SJ42					30
SJ43					35
SJ44					40
SJ45					46.5

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
