# Peer review of "Ultrasonic Detection Method for Grouted Defects in Grouted Splice Sleeve Connector Based on Wavelet Pack Energy"

_sensors, 2019, doi:10.3390/s19071642_

Reviewer 1 Report

The paper presents an ultrasonic method for grouted defect detection in grouted splice sleeves making use of wavelet pack energy approach. Increased efficiency of the method is shown with reference to traditional ultrasonic methods. The limitations of classical tools can be overcome via the application of wavelet packet analysis applied for signal processing in the developed detection system. A grouting defect index is proposed based on wavelets. Both theoretical and experimental works are presented providing with interesting properties of the proposed approach. Although the proposed technique refers to the already known measurement technique, the work is worth to be published due to its proven characteristics. However some corrections and merit comments are required before the manuscript is recommended for publication. Moreover many flaws appear, and the paper needs language improvements.

It is necessary to comment on the influence of the environmental conditions (temperature and humidity) on the quality of the measured data and efficiency of detection procedure.

Could you please comment of the smallest damage that could be detected. Maybe more physical explanation regarding the obtained results would help to answer this question.

It is stated that the frequency sub-bands are adaptively selected to match the frequency spectrum of the signal. What is the relationship between the widths of these sub-bands and the signal spectrum?

How to decide which sub-band is the most efficient for the presented procedure in a real case - when no initial data is provided?

In the checked PDF-file of the manuscript there is a lack of the photograph in section d) SJ4 group in Figure 2.

Many flaws and syntax errors, e.g.:  

Page 1 row 14: shouldn’t be?   “affecting” -> “affected”

p.1  r.15 “traditional ultrasonic method is difficult to detect” -> “it is difficult to use traditional ultrasonic method to detect”

p.1 r 25: “advantages. Well control” -> “advantages: effective control”

p1 r27: “In-site” -> “On-site”

p1 r29: “Precast concrete” -> “A precast concrete”

p1 r33: “grouting material need” -> “grouting material needs”

… definitely needs farther language improvement

Author Response

Dear reviewers and editor,

Thank you very much for the constructive comments and suggestions. These comments are valuable and very helpful for improving this paper. The paper has been carefully revised according to them. The texts with GREEN label mean the modified contents, and the texts with BLUE label are the added contents.

Reviewer 1:

1.1. It is necessary to comment on the influence of the environmental conditions (temperature and humidity) on the quality of the measured data and efficiency of detection procedure.

Answer: Thanks for the Reviewer’s valuable suggestion. From the references [1-3], it can be known that if the temperature changes in a wide range, the influence of temperature on the velocity and amplitude of the ultrasonic wave propagation cannot be ignored. Furthermore, from the references [4-5], it can be known that the humidity also has influence on the speed of the ultrasonic wave.

However, the specimens in this paper were placed in the laboratory, and the detection time was from 10 o 'clock to 18 o 'clock in summer. The temperature and humidity in the laboratory changed very little, so the temperature and humidity in the laboratory caused very little change of ultrasonic wave, which had little influence on the quality of the measurement data and the efficiency of detection procedure.

Since the main purpose of this paper is to verify the effectiveness of the proposed method, comments on the environmental conditions won’t be added to this paper.

Thanks again for the valuable comments. In order to better apply the proposed method in practice, the influence of environmental conditions needs to be studied in depth.

[1] Francesco, L.D.S.; Salamone S. Temperature effects in ultrasonic Lamb wave structural health monitoring systems. Journal of the Acoustical Society of America. 2008, 124, 161-74.

[2] Putkis, O.; Dalton, R.P.; Croxford, A.J. The influence of temperature variations on ultrasonic guided waves in anisotropic CFRP plates. Ultrasonics. 2015, 60, 109-116.

[3] Ji, J.; Li, H.; Yan, G. Effect of temperature on ultrasonic wave velocity in concrete. Journal of Building Materials. 2008, 11, 349-352.

[4] Berriman, J.; Purnell, P.; Hutchins, D.A.; Neild, A. Humidity and aggregate content correction factors for air-coupled ultrasonic evaluation of concrete. Ultrasonics. 2005, 43, 211-217.

[5] El-Sabbagh, A.; Steuernagel, L.; Ziegmann, G. Ultrasonic testing of natural fibre polymer composites: effect of fibre content, humidity, stress on sound speed and comparison to glass fibre polymer composites. Polymer Bulletin. 2013, 70, 371-390.

1.2. Could you please comment of the smallest damage that could be detected. Maybe more physical explanation regarding the obtained results would help to answer this question.

Answer: Thanks for the Reviewer’s valuable suggestion. More physical explanation is added to a paragraph in page No.10. A part of the revised paragraph is shown as follow.

When the ultrasonic wave encountered grouted defects in the propagation, diffraction, and scattering occurred, and the energy changed. From the perspective of energy variation, it could be seen thatOnly the situation onlywhenthegrouting defectgrouted defectreachedes a certain widthsizewoulditwil cause the obvious changes of theenergy ofultrasonic detection signal, and the energy proportion in different frequency sub-bands hadve asignificant difference, which iswaspresented as the difference of the WPEVs betweenthespecimen without grouting defectgrouted defectsandthespecimens with grouting defectgrouted defects.

1.3. It is stated that the frequency sub-bands are adaptively selected to match the frequency spectrum of the signal. What is the relationship between the widths of these sub-bands and the signal spectrum?

Answer: The authors are sorry for the wrong expression. Several sentences in page No.3 misrepresented what we wanted to represent. These sentences have been modified and they are shown as follow.

However, the wavelet packet analysis decomposes both the high frequency part and the low frequency part according to the characteristics of an ultrasonic detection signal, adaptively selects the corresponding frequency sub-bands, and distributes different signal components to different frequency sub-bands.

Besides, there is no relationship between the width of these sub-bands and signal spectrum. The width of the sub-bands W is determined by the sampling rate of the detection signal SR and the decomposition layers N. According to the Nyquist sampling theorem, their relationship is shown as following formula.

1.4. How to decide which sub-band is the most efficient for the presented procedure in a real case - when no initial data is provided?

Answer: In this paper, the excitation signal is a five-peaks sinusoidal signal modulated by the hanning window, of which the main frequency is 50kHz. The main frequency range of the excitation signal is 24.42-73.26kHz. It means that the signal energy is obviously concentrated between 24.42-73.26kHz. Therefore, the corresponding sub-bands, namely sub-bands 3-6, were selected in the processing of experimental data. The signal energy basically varies in these sub-bands.

Thus, if no initial data are provided, several sub-bands where the energy is concentrated in can be selected as effective sub-bands for further calculation.

1.5. In the checked PDF-file of the manuscript there is a lack of the photograph in section d) SJ4 group in Figure 2.

Answer:Thank you very much. The figure is added in page No.7.

a) SJ1 group

b) SJ2 group

c) SJ3 group

d) SJ4 group

Figure 2 The physical diagram of the concrete specimens

1.6. Many flaws and syntax errors, e.g.: 

Page 1 row 14: should not be?  “affecting” -> “affected”

p.1 r.15 “traditional ultrasonic method is difficult to detect” -> “it is difficult to use traditional ultrasonic method to detect”

p.1 r.25: “advantages. Well control” -> “advantages: effective control”

p.1 r.27: “In-site” -> “On-site”

p.1 r.29: “Precast concrete” -> “A precast concrete”

p.1 r.33: “grouting material need” -> “grouting material needs”

 Answer: The authors are very sorry for the incorrect writing. Based on Reviewer’s comment, the paper has been carefully and thoroughly proofread and revised. Many syntactic and grammatical mistakes have been revised (some corrections are shown as following table 1 ).

Table 1. The main modifications in the paper

Original

Modified

Well control

Effective control

traditional ultrasonic   method is difficult to detect

it is difficult to   identify grouting defects in the GSS connector effectively using the   traditional ultrasonic parameters.

In-site

On-site

Precast concrete

A precast concrete

grouting material need

grouting material needs

difficulties in   interpretation

difficulties in the   interpretation

based on wavelet packet analysis

based on the wavelet   packet analysis

which has a great hope   of achieve the goal of

Which has significant   potential in

grouting defects in GSS   connector

grouting defects in a   GSS connector

an extension of Fourier   transform

an extension of the Fourier   transform

in high frequency

at high frequency

Reviewer 2 Report

Comments on paper submitted to Sensors

 Ultrasonic detection method for grouted defect in grouted splice sleeve connector based on wavelet pack energy

Manuscript ID Sensors-453314

 General comments

The paper shows the a method to detect defects in precast concrete elements. The presented work could be of interest to researchers and practicing engineers, but is currently not presented in a form that allows applications.

 The abstract is poorly written, does not show the focus of the study, does not describe the results clearly, and lacks a conclusion.

 The introduction does not explain the scope and limitations of the work.

 The literature review is insufficient, presented as an annotated bibliography, and presented at different places in the manuscript.

 The writing of the paper is too poor to judge the methods. It’s not clear which work is taken from the literature, and what is the new contribution of the authors.

 The analysis of the results seems to be limited to reporting observations.

 Recommendations for practice are missing.

Future work is missing.

 The conclusions are limited to reporting observations.

 The paper is very poorly written. The poor writing makes it very difficult to read this paper. It requires a complete rewrite.

 Reference to figures and tables has been done correctly. The figures and tables are of poor quality.

 A list of notations is missing.

 To conclude: Evaluating the technical contents of this paper is nearly impossible, since the writing is very poor. The authors seem to focus only on reporting their work, without framing it within the literature, and current practice, and without deeper analyzing their results.

Technical/Editorial comments

Please find here the detailed technical (T) and editorial (E) comments[1]:

Page

Line

T/E

Comment

1

21

T

Abstract needs a complete rewrite.

6

156

T

Figure1a seems to be wrong, the   dimension of 425 mm is indicated over the wrong range.

7

158

T

Figure 2d is missing.

[1] Pages in PDF

Author Response

Dear reviewers and editor,

Thank you very much for the constructive comments and suggestions. These comments are valuable and very helpful for improving this paper. The paper has been carefully revised according to them. The texts with GREEN label mean the modified contents, and the texts with BLUE label are the added contents.

Reviewer 2:

2.1. The abstract is poorly written, does not show the focus of the study, does not describe the results clearly, and lacks a conclusion.

Answer: In response to the comment, the abstract has been rewritten, and it is shown as follow.

Abstract: Grouted splice sleeve (GSS) connectors are mainly used in precast concrete structures. However, errors in manual operation during construction lead to grouteding defects intheGSS connector. As a result, the overall mechanical propertypropertiesof the structure isareseriouslynegativelyaffectingaffected. DueOwingto the complex combinations of precast concrete elementsmemberswith GSS connector,it is difficult to detectthegrouteddefect effectivelyusingtraditional ultrasonicacousticparametersmethod is difficult to detect grouting defects effectively. Inthis paper, the ultrasonic method andawavelet packet analysis algorithmwaserecombineddevelopedorder to effectively detect grouteding defectsusing the ultrasonic method,. Firstly,an ultrasonic detection method based on wavelet packet energy iswasproposed firstly. andtheagrouteddefect index (GDI)wasgivenproposedtojudgedeterminethe degree of grouteddefects.Secondly, the correspondinga simple and convenientultrasonic detectingmeasurementsystemand a detecting operationwereis developed, and grouting defect indexes (GDI) are given to judge the degree of grouting defects..Adetection proceduresuitablefor the groutingeddefects is proposed.Finally, severalfourgroups of concrete specimens with GSS connectors arewereconstructed to verify the effectiveness of the method., In each group,and five sizes of grouteddefects in each grouparewereconsideredtested.especially the specimens with different defect widths. The results demonstratedthatwhen the grouteddefect reacheda certain size,theproposedmethod cancoulddetect grouteding defects effectively.The proposed method is effective, simple instrument, which iandseasy to implemented atconstruction sitewith simple instrument,whichprovidesingan innovativemethod for the detection of grouteddefects of precast concrete members.

 2.2. The introduction does not explain the scope and limitations of the work.

 Answer: In order to explain the scope and limitations of the work, several new sentences and two references are added to the introduction in page No.2. They are shown as follow.

Precast concrete structureshashavemany advantageshave received increasing research interestsrecently, especially in a global trend ofoff-sitemodular construction.Their advantages include:WellEeffectivecontrol over the quality of materials and workmanship, low resource consumption and cost efficiency,Further, besides,theIOon-site construction/assemblymodeof precast concrete structureshas characteristicsis characterizedbybenefits fromtheof small lowimpact of weather conditions and low labor demand [1-3]. Precast concrete structuresmeets the development requirements of construction, industrialization,and urbanization.The most important part used to connect precast concrete elementsmembersis calledthegrouted splice sleeve (GSS) connector,which is composed ofasplice sleeve, connectingasteel rebar and high-strength, micro-expansion, cement-based grouting materials. In order to ensure the continuity of load transfer, grouting materialsneedsto be regardedviewedasahigh performance bonding materialsfor connecting steel rebar [4].InFortheion-site constructionstagemode, humanvariousfactorsmayand construction technologylead to the failure of grouting materialsstrength,theinadequate sealing of grouting joints ortheobstruction of grouting inlets and outlets. These deficienciescause the emergence of groutingeddefects. A grouted defect in a splice sleeve has a significant negative impact on the connection quality and the load-transmitting performance of components, which may lead to insufficient anchorage length of the connecting rebar, a reduction in the bearing capacity and seismic performance of the precast structure. These results make the precast structure unable to achieve the same performance as that of the cast-in-place structure.

Because ofOwing tothe complex combinations of precast concrete elementsmemberswithaGSS connector and the concealment of grouting material insidethesplice sleeve, it is quite difficult to detect groutingeddefects insidethesplice sleeve. PresentCurrentresearches mainly focus onthebond anchorage mechanism [3, 45,6], joint mechanical properties [5,67,8],and deformation or damage monitoring [7,89,10], but there are fewer researchesthere is limitedresearchonabout grouting defectgrouted defects detection intheGSS connector.Gao [11] used industrial CT to detect concrete specimens with a GSS connector in the laboratory, and the result showed that industrial CT could effectively show the grouting defectgrouted defects in a GSS connector. Zhou [12] used industrial CT to detect GSS connectors with foam particle defects in the laboratory, and the result showed that industrial CT could determine the location of defects and perform a quantitative analysis of the defect size. However, due to the high hardware requirements, high cost, and harsh application conditions, industrial CT instruments are currently limited to laboratory application, as it is difficult to use the instruments at the construction site.

11. Gao, R.; Li, X.; Zhang, F.; Xu Q.; Wang, Z.; Liu, H. Experiment on detection of coupler grouting compactness based on industrial X-CT. Nondestructive Testing. 2017, 04, 12-17.

12. Zheng, Z.; Xiao, Y.; Li, D.; Tian, Y.; Xie, X.; Zhang, T. Research on inspection method of sleeve grouting connector based on CT technology. Construction Technology. 2018, 47, 69-74.

 2.3. The literature review is insufficient, presented as an annotated bibliography, and presented at different places in the manuscript.

 Answer: The authors sincerely appreciate Reviewer’s valuable comment. In responding to the comment, two literature are added to the introduction in page No.2, as shown in question 2.2.

Besides, the reference number was in wrong order before. We have renumbered the references so they appear in sequential numerical order.

2.4. The writing of the paper is too poor to judge the methods. It’s not clear which work is taken from the literature, and what is the new contribution of the authors.

 Answer: The authors are very sorry for our poor writing. The algorithm of wavelet packet analysis is taken from the literature. However, the authors proposed a new detection method for the grouting defect using the existing signal processing technique, containing the definition of the grouting defects index ,and the development a simple and convenient detection system and a detection operation, which are the new contribution of the authors.

 In order to make it clear which work is taken from the literature, the wrong grammars have been corrected carefully and three references as follow are added in section 2.1 in page No.4.

27. Mallat, SG. A theory for multiresolution signal decomposition: the wavelet representation. IEEE Transactions on Pattern Analysis and Machine Intelligence. 1989, 7, 674-693.

28. Mallat, SG. Multiresolution approximations and wavelet orthonormal bases of L2(R). Transactions of the American Mathematical Society. 1989, 315, 69-87.

29. Mallat, SG. Multifrequency channel decompositions of images and wavelet models. IEEE Trans. Acoustics, Speech, and Signal Processing. 1989, 37, 2091-2110.

 2.5. The analysis of the results seems to be limited to reporting observations.

 Answer: In order to enrich the analysis of the results, a few sentences are added to a paragraph in page No.10. The revised paragraph is shown as follow.

When the ultrasonic wave encountered grouting defects in the propagation, diffraction, and scattering occurred, and energy changed. From the perspective of energy variation, it could be seen that only Only the situationwhenthegrouting defect reachedes a certain widthsize,wilwoulditcause the obvious changes of theenergy ofultrasonic detection signal, and the energy proportion in different frequency sub-bands hadve asignificant difference, which iswaspresented as the difference of the WPEVs betweenthespecimen without grouting defectsandthespecimens with grouting defects. Therefore, the grouting defect index (GDI) was used to evaluate the energy level of ultrasonic detection signal of specimens with different sizes of grouting defects, so as to identify the grouting defects.

2.6. Recommendations for practice are missing.

 Answer: The authors have added recommendations for practice in page No.15 according to the Reviewer’s comments. And they are shown as follow.

In practice, several typical precast concrete members’ models can be prepared in the laboratory according to the construction process and materials at the construction site. The grouting operation is carried out carefully according to the requirement to ensure that there are no grouted defect in models. The models are detected according to the proposed method, and the WPEVs and GDIs are calculated to obtain the grouted defect identification baseline. Meanwhile, the precast concrete members at the construction site are also detected, and then the GDIs of members are calculated based on the WPEVs obtained from the laboratory models. Finally, compared with the obtained identification baseline, if the GDI of a member exceeds the line, it can be regarded that the grouted defect exists in the member.

 2.7. Future work is missing.

 Answer: Considering the Reviewer’s suggestion, we have added future work in page No.16. And they are shown as follow.

In order to better promote the method in engineering practice, follow-up research can be conducted from the following aspects.

I.         There are transverse rebars in precast concrete members, which will affect the detection. The influence of transverse rebars can be further studied.

II.      The environmental factors, such as temperature and humidity, are different in different areas. The influence of environmental factors can be further studied to adapt the method to the construction sites located in various areas.

 2.8. The conclusions are limited to reporting observations.

 Answer: In order to enrich the conclusions, a paragraph is added to the conclusion in page No.16. “The proposed ultrasonic detection method based on wavelet pack energy achieved effective detection of grouted defects. Moreover, it has low instrument requirements, simple application conditions, and it is easy to apply at the construction site, which provides good potential in the detection of grouted defects of precast concrete members.

 2.9. Reference to figures and tables has been done correctly. The figures and tables are of poor quality.

 Answer: In response to the comment, two figures of poor quality in page No. 8 are redrawn. And they are shown as follow.

 Figure 3. The schematic diagram of the detecting system.

Figure 4. The physical diagram of the detecting system.

 2.10. A list of notations is missing.

 Answer: A list of notations are added in page No.16. It is shown as follow.

 List of notations

Mother wavelet function

Wavelet subspace

Scaling space

Scale function

Wavelet function

Decomposition of ultrasonic detection signal S

Energy of decomposition

Wavelet packet energy of the signal S

Wavelet packet energy vector of signal S

GDI

Grouting defect index

2.11. Figure 1a seems to be wrong, the dimension of 425 mm is indicated over the wrong range. Figure 2d is missing.

 Answer: Thank you very much. The Figure 1a is modified and the Figure 2d is added. The modified Figure 1a is shown as follow and the Figure 2d is as shown in question 1.5.

a) SJ1 group

b) SJ2 group

c) SJ3 group

d) SJ4 group

Figure 1 The schematic diagram oftheconcrete   specimens

Round  2

Reviewer 1 Report

Even though the Authors have improved the paper significantly, it still looks like a technical report presenting the experimental results. Before the paper may be considered for publication, more physical explanation should be added to the work, especially to the last part of the paper, including the last section.

Moreover there are still some language flaws and the paper requires check.

Author Response

Dear reviewers and editor,

Thank you very much for the constructive comments and suggestions. These comments are valuable and very helpful for improving this paper. The paper has been carefully revised according to them. The texts with BLUE label are the added contents.

 Reviewer 1:

1.1. Even though the authors have improved the paper significantly, it still looks like a technical report presenting the experimental results. Before the paper may be considered for publication, more physical explanation should be added to the work, especially to the last part of the paper, including the last section.

 Answer: Thanks for the Reviewer’s valuable suggestion. Authors have tried their best to add more physical explanation to the paper. In this paper, the influence of grouted defect on ultrasound can be visually reflected in the WPEV of ultrasonic detection signal, thus, most of the physical explanation concentrated in this section. The GDI in the last section is calculated by the WPEV and is used to judge the existence of grouted defects. Therefore, there is only a little physical explanation. More physical explanation has been added to the paper, and it is shown as follows.

In page No.5, a sentence is added. “In this paper,thewavelet packet analysis was used as a signal processing tool to analyze the ultrasonic detection signal of precast concrete members with a GSS connector, and an evaluation index based on the wavelet packet energy was proposed and employed to detect the grouted defect in the GSS connector.

In page No.7, a sentence is added. “In this paper, an evaluation index is defined based on the ultrasonic detection signal processed by the wavelet packet analysis. Suppose that Aan ultrasonic detection signal S is decomposed by an N-level wavelet packet decomposition into a 2N signal set {X1, X2, X3, …, X2N} with

Moreover, several sentences are added to section 3.2 in page No.7 to explain the processing details of ultrasonic detection signal. “Symlets wavelet base sym14 was regarded as the mother wavelet. The frequency band was not overlapped because of the orthogonality of Symlets wavelet base. The ultrasonic detection signals of the specimens were decomposed into ten levels. The wavelet packet coefficients of the 10th level were extracted to calculate the energy proportion of frequency sub-bands 3, 4, 5 and 6 (the frequency range was from 24.42 to 73.26 kHz), and then the WPEVs of the detection signals were built. To reduce the influence of the operation error, the average of the three groups’ WPEVs at each point was taken as its final WPEV. Finally, the GDIs of each specimen were calculated using the proposed equation 16.

In page No.12 of the revised manuscript, “Analysis of frequency sub-bands indicated that the grouted defect in a GSS connector could have a significant impact on the signal energy distribution. When the ultrasonic wave encountered grouted defects in the propagation, diffraction, and scattering occurred, and the energy mainly changed at frequency sub-band 4 and 5. From the perspective of energy variation, it could be seen that only when the grouted defects reached certain sizes would it cause the obvious change of the energy of ultrasonic detection signal, and the energy proportion in different frequency sub-bands had a significant difference, which was presented as the difference of the WPEVs between the specimen without grouted defects and the specimens with grouted defects. Consequently, it could be concluded that the WPEV of a detection signal could effectively reflect the grouted defect in a GSS connector.

In page No.16, a sentence is added. “From the point of energy variation, the proposed grouted defect index (GDI) was used to evaluate the energy level of ultrasonic detection signal of specimens with different sizes of grouted defects, so as to identify the grouted defects.

In page No.17 of the revised manuscript, “Owing to the influence of concrete aggregates distribution, the WPEVs of the ultrasonic detection signals of different points on a specimen were different, and this led to its GDIs fluctuating within a certain range. As shown in Figure 10, the GDI of each specimen without grouted defects was less than 0.14. Thus, the grouted defect identification baseline in the experiment was 0.14, which meant that when all the GDIs of a specimen with certain size of groutinggrouteddefect was more than 0.14, it was considered that this grouted defect could be effectively distinguished by the GDI.”

1.2. Moreover there are still some language flaws and the paper requires check.

 Answer: Thanks for the Reviewer’s comment. The paper has been edited by an English editing service provided by MDPI. And it also has been carefully and thoroughly proofread and revised by authors. Many syntactic and grammatical mistakes have been revised (some corrections are shown in table 1).

Table 1. The main modifications in the paper

Original

Modified

with GSS   connector

with a GSS   connector

with simple   instrument

with simple   instruments

the complex combinations   of

the complex   structure of

a wavelet   packet analysis

the wavelet   packet analysis

the ultrasonic   detecting system

an ultrasonic   detecting system

wavelet packet   analysis

the wavelet   packet analysis

the excitation   signal

an excitation   signal

grouted defect   index

a grouted   defect index

water based   polymer gel ultrasound coupler

water-based   polymer gel ultrasound coupler

the probe

the probes

wouldl

would

incidents

incidented

affects

affected

propagates

propagated

will

would

are

were

SJ11

the SJ11

grouting defect

grouted defect

Reviewer 2 Report

This work remains poorly written - the overall structure of the work has not been improved.

Author Response

Dear reviewers and editor,

Thank you very much for the constructive comments and suggestions. These comments are valuable and very helpful for improving this paper. The paper has been carefully revised according to them. The texts with BLUE label are the added contents.

 Reviewer 2:

2.1. This work remains poorly written - the overall structure of the work has not been improved.

 Answer: Thanks for the Reviewer’s comment. The authors have tried their best to improve the overall structure of the work.

Firstly, the language flaws in the paper have been corrected. The paper has been edited by an English editing service provided by MDPI. And it also has been carefully and thoroughly proofread and revised by authors. Many syntactic and grammatical mistakes have been revised (some corrections are shown in table 1).

Table 1. The main modifications in the paper

Original

Modified

with GSS   connector

with a GSS   connector

with simple   instrument

with simple   instruments

the complex   combinations of

the complex   structure of

a wavelet   packet analysis

the wavelet   packet analysis

the ultrasonic   detecting system

an ultrasonic   detecting system

wavelet packet   analysis

the wavelet   packet analysis

the excitation   signal

an excitation   signal

grouted defect   index

a grouted   defect index

water based   polymer gel ultrasound coupler

water-based   polymer gel ultrasound coupler

the probe

the probes

wouldl

would

incidents

incidented

affects

affected

propagates

propagated

will

would

are

were

SJ11

the SJ11

grouting defect

grouted defect

  Secondly, the abstract has been revised carefully. It is shown as follows.

Abstract: Grouted splice sleeve (GSS) connectors are mainly used in precast concrete structures. However, errors in manual operation during constructioncauselead to grouted defects in the GSS connector,which will lead toanegative effect of. As a result,the overall mechanical properties of the structuresisare seriously negatively affectingaffected. Owing to the complexstructurecombinations of precast concrete members withaGSS connector, it is difficult to detectthe grouted defectseffectively using traditional ultrasonic parameters. In this paper, a wavelet packet analysis algorithm was developed to effectively detect grouted defects using the ultrasonic methodand averified experiment was carried out. Firstly, a grouted defect index (GDI) was proposed to determine the degree of grouted defects. Secondly, a simple and convenient ultrasonic measurement system were developed, and a. A detection procedure suitable for the grouted defectswasis proposed. Finally, four groups of concrete specimens withaGSS connectors weremanufacturedconstructed to verify the effectiveness of the method, then.iIn each group, five sizes of grouted defects weredetectedtested. The results demonstrated that when thethe grouted defects reached certain sizes, the proposed method could detectthegrouted defects effectively.In this paper, a wavelet packet analysis algorithm was developed to effectively detect grouted defects using the ultrasonic method, and a verified experiment was carried out. Laboratory detection was performed on the concrete specimens with a GSS connectorbefore grouting, in which the grouted defects were mimicked by setting EPE with five sizes in five GSS connectors of each specimen group. A simple and convenient ultrasonic detection system was developed and the specimens were detected. According to the proposed grouted defect index, the results demonstrated that when the grouted defects reached certain sizes, the proposed method could detect the grouted defects effectively.The proposed method is effective, and easy to implement at a construction site with simple instruments, which provides an innovative method forthe detection of grouted defectsdetectionof precast concrete members.

 Thirdly, the whole section 3.1. Design and manufacture of concrete specimens with a GSS connector in last manuscript is adjusted to be the section 2.Fabrication of concrete specimens with a GSS connectorin new manuscript. And the whole section 3.2 Defect detecting system and detecting points layout in last manuscript is adjusted to be the section 4. Experimental development in new manuscript. The structure of the new manuscript is as follows.

1. Introduction

2. Fabrication of concrete specimens with a GSS connector

3. 2.Methodology

3.1. Wavelet packet analysis

23.2. Proposed Ggrouted defect index

4. 3.Experimental development

5. 4.Experimental results and analysis

5.1. The WPEVs ofan ultrasonic detection signalsof detecting points

5.2. Grouted defects detection using wavelet packet analysis based indexGDI of detecting points

6. Conclusion

 Fourthly, some sentences are added to the paper to make the expression clearer, and they are shown in follow.

In page No.5, a sentence is added. “In this paper,thewavelet packet analysis was used as a signal processing tool to analyze the ultrasonic detection signal of precast concrete members with a GSS connector, and an evaluation index based on the wavelet packet energy was proposed and employed to detect the grouted defect in the GSS connector.

In page No.7, a sentence is added. “In this paper, an evaluation index is defined based on the ultrasonic detection signal processed by the wavelet packet analysis. Suppose that Aan ultrasonic detection signal S is decomposed by an N-level wavelet packet decomposition into a 2N signal set {X1, X2, X3, …, X2N} with

Moreover, several sentences are added to section 3.2 in page No.7 to explain the processing details of ultrasonic detection signal. “Symlets wavelet base sym14 was regarded as the mother wavelet. The frequency band was not overlapped because of the orthogonality of Symlets wavelet base. The ultrasonic detection signals of the specimens were decomposed into ten levels. The wavelet packet coefficients of the 10th level were extracted to calculate the energy proportion of frequency sub-bands 3, 4, 5 and 6 (the frequency range was from 24.42 to 73.26 kHz), and then the WPEVs of the detection signals were built. To reduce the influence of the operation error, the average of the three groups’ WPEVs at each point was taken as its final WPEV. Finally, the GDIs of each specimen were calculated using the proposed equation 16.

In page No.16, a sentence is added. “From the point of energy variation, the proposed grouted defect index (GDI) was used to evaluate the energy level of ultrasonic detection signal of specimens with different sizes of grouted defects, so as to identify the grouted defects.

 Fifthly, more physical explanation is added to the paper to explain the influence of the grouted defect on ultrasound. It is added to page No.12 and page No.17. It is shown as follows.

Analysis of frequency sub-bands indicated that the grouted defect in a GSS connector could have a significant impact on the signal energy distribution. When the ultrasonic wave encountered grouted defects in the propagation, diffraction, and scattering occurred, and the energy mainly changed at frequency sub-band 4 and 5. From the perspective of energy variation, it could be seen that only when the grouted defects reached certain sizes would it cause the obvious change of the energy of ultrasonic detection signal, and the energy proportion in different frequency sub-bands had a significant difference, which was presented as the difference of the WPEVs between the specimen without grouted defects and the specimens with grouted defects. Consequently, it could be concluded that the WPEV of a detection signal could effectively reflect the grouted defect in a GSS connector.

Owing to the influence of concrete aggregates distribution, the WPEVs of the ultrasonic detection signals of different points on a specimen were different, and this led to its GDIs fluctuating within a certain range. As shown in Figure 10, the GDI of each specimen without grouted defects was less than 0.14. Thus, the grouted defect identification baseline in the experiment was 0.14, which meant that when all the GDIs of a specimen with certain size of groutinggrouteddefect was more than 0.14, it was considered that this grouted defect could be effectively distinguished by the GDI.

Round  3

Reviewer 2 Report

Whereas the grammar and punctuation of the writing has been improved, the overall style remains poor:

- the introduction does not concisely answer the question: why do we care about this research?

- style of literature review is not good - presented as annotated bibliography

- future work should be in a discussion section but is in the conclusion

- etc

Author Response

1.1. The introduction does not concisely answer the question: why do we care about this research

 Answer: Thanks. A part of the introduction has been improved to concisely answer the question, and it is shown as follow.

Precast concrete structures have received increasing research interests recently, especially in a global trend of off-site modular construction. Their advantages include: effective control over the quality of materials and workmanship, low resource consumption and cost efficiency, besides, the on-site construction/assembly of precast concrete structures benefits from the low impact of weather conditions and low labor demand [1-3]. The most important part used to connect precast concrete members is called the grouted splice sleeve (GSS) connector, which is composed of a splice sleeve, connecting a steel rebar and high-strength, micro-expansion, cement-based grouting materials. In order to ensure the continuity of load transfer, grouting materials need to be viewed as high performance bonding materials for connecting steel rebar [4]. In the on-site construction stage, various factors may lead to the failure of grouting materials, the inadequate sealing of grouting joints or the obstruction of grouting inlets and outlets. These cause the emergence of grouted defects. Grouted defects in a splice sleeve has a significant negative impact on the connection quality and the load-transfer performance of members, which may lead to insufficient anchorage length of the connecting rebar, a reduction in the bearing capacity and seismic performance of the precast structure. These results make the precast structure unable to achieve the same performance as that of the cast-in-place structure. The grouted defect in a GSS connector remains a significant concern, because the grouted defect due to grouting operation mistakes can negatively affect load transfer and decrease the bearing capacity and seismic performance of the precast structures.

Owing to the complex structure of precast concrete members with a GSS connector and the concealment of grouting material inside the splice sleeve, it is quite difficult to detect grouted defects inside the splice sleeve. Current researches mainly focus on the bond anchorage mechanism [5,6], joint mechanical properties [7,8], and deformation or damage monitoring [9,10], but there is limited research on the detection of grouted defects in the GSS connector. Gao [11]usedindustrialCT to detect concrete specimens with a GSS connector in the laboratory, and the result showed that industrial CT could effectively show the grouted defects in the GSS connectors.Zhou [12] usedindustrial CT to detect GSS connectors with foam particle defects in the laboratory, and the result showed that industrial CT could determine the location of defects and perform a quantitative analysis of the defect size. However,Although the industrial CT can achieve the detection of grouted defects in a GSS connector in the laboratory [11,12], due to the high hardware requirements, high cost, and harsh application conditions, industrial CT instruments are currently limited to laboratory application. Therefore, it is a challenging research topic to develop a grouted defects detection method which is reliable and easy to implement on site.

1.2. The style of literature review is not good – presented as annotated bibliography.

Answer: Thanks. Two of the literatures in page No.2 were summarized, and it is shown as follow.

Although the industrial CT can achieve the detection of grouted defects in a GSS connector in the laboratory [11,12], due to the high hardware requirements, high cost, and harsh application conditions, industrial CT instruments are currently limited to laboratory application.”

What’s more, the style of literature review in page No.2 is changed, and it is shown as follow.

“With the development of signal processing technology, more powerful parameters of ultrasonic detection have been developed for the defect detection from the energy perspective [22]. In [23], a structural health monitoring method based on the wavelet packet was proposed, and the effective monitoring of the generation and development process of cracks under different damage states of members by using piezoelectric aggregates was established. In [24], a method based on the energy spectrum analysis of the piezoelectric ceramic stress wave wavelet packet to identify the interfacial bond performance of a concrete-filled steel tubular column was proposed, and the result showed that the method could achieve relatively ideal detection. In [25], the piezoelectric aggregate excitation stress wave was used to detect the damage of a simulated concrete beam. These results showed that the damage index based on the wavelet packet analysis was very sensitive to the damage of members within a certain propagation distance and it could effectively reflect the damage degree of members. Therefore, the wavelet packet analysis algorithm, which has significant potential in the detection of grouted defects in a GSS connector, can be adopted to analyze the ultrasonic detection signal of precast concrete members.”

 1.3. Future work should be in a discussion section but is in the conclusion.

 Answer: Thanks. Future work becomes a new section in the revised manuscript. And it is shown as follow.

7.      Future work

In order to better promote the method in engineering practice, follow-up research can be conducted from the following aspects.

I.          There are transverse rebars in precast concrete members, which will affect the detection. The influence of transverse rebars can be further studied.

II.       The environmental factors, such as temperature and humidity, are different in different areas. The influence of environmental factors can be further studied to adapt the method to the construction sites located in various areas.

 1.4. Besides, a part of the conclusion has been improved by the authors, and it is shown as follow.

In this paper, an ultrasonic detection method based on wavelet packet energy for grouted defects in a GSS connector of the precast concrete structure and an index to judge the existence of the grouted defects was proposed. The effectiveness of the proposed method was experimentally verified with a simple and convenient detection system and four groups of concrete specimens with GSS connectors. Based on the above results, the following conclusions can be made.
